# Predictors of Deterioration in Mental Well-Being and Quality of Life among Family Caregivers and Older People with Long-Term Care Needs during the COVID-19 Pandemic

**DOI:** 10.3390/healthcare12030383

**Published:** 2024-02-01

**Authors:** Paolo Fabbietti, Sara Santini, Flavia Piccinini, Cinzia Giammarchi, Giovanni Lamura

**Affiliations:** 1Center for Biostatistic and Applied Geriatric Clinical Epidemiology, IRCCS INRCA-National Institute of Health and Science on Aging, 60124 Ancona, Italy; p.fabbietti@inrca.it; 2Centre for Socio-Economic Research on Aging, IRCCS INRCA-National Institute of Health and Science on Aging, 60124 Ancona, Italy; f.piccinini@inrca.it (F.P.); g.lamura@inrca.it (G.L.); 3Scientific Direction, IRCCS INRCA-National Institute of Health and Science on Aging, 60124 Ancona, Italy; c.giammarchi@inrca.it

**Keywords:** COVID-19 pandemic, informal caregivers, mental well-being, older people, quality of life, long-term care

## Abstract

Background: During the COVID-19 pandemic, reduced access to care services and fear of infection prompted families to increase home care for their older relatives with long-term care needs. This had negative effects on both members of the caring dyad, impacting their quality of life (QoL) and mental well-being. This study investigated the factors that influenced the mental well-being and QoL of 239 dyads, before and after the first pandemic wave in Italy. Methods: Data were collected through a survey on the use of health and social care services and interventions by older care recipients living in the community and their family caregivers. Factors associated with deterioration of mental well-being and QoL in older care recipients (mean age 86.1 years old) and their family caregivers after the pandemic were studied. Results: The importance attached by family caregivers to the skills and training of healthcare professionals was a protective factor against the deterioration in the well-being of older care recipients. Similarly, the importance associated by family caregivers to the help received from healthcare professionals was a protective factor for QoL. Financial hardship of older care recipients was a risk factor for deterioration in caregivers’ mental well-being, while support from other family members was a protective factor for QoL. Conclusions: The presence of attentive healthcare professionals, a supportive family environment, and economic support can reduce the burden on both the caregiver and the older care recipient. These aspects need to be considered in any future emergency situation and when planning care services for community-dwelling older people.

## 1. Introduction

In Europe, more than a fifth of the population is over 65 years old [1]. The progressive ageing of the population is leading to a staggering increase in the proportion of older individuals with multiple health issues. Multimorbidity, defined as the presence of two or more chronic conditions in the same individual [2], is associated with reduced functional ability [3,4,5], poorer quality of life (QoL) [6], and higher healthcare utilization [7], thus increasing the need for long-term care, especially in later life [8,9]. Moreover, the presence of two or more chronic conditions appears to have a greater and cumulative effect on an individual’s general health in the long term than in the short one [10].

The coexistence of physical and mental health conditions is associated with a significant decrease in health-related QoL (HRQoL) [11].

Irrespective of the level of development of welfare state services and long-term care systems [12], the greatest burden of caring for older individuals with long-term care needs falls on their closest relatives [13]. This burden is greater in countries with a family-based care system, where care the older people is provided primarily at home, due to the scarcity of residential services and the desire of older people to age in place and live in their own communities for as long as possible [14]. However, this goal, which is also one of the key objectives at the top of the European Union’s political agenda, risks worsening the QoL of informal caregivers, who provide unpaid care to older family members, if they are not given sufficient and appropriate support. Many studies highlight the negative impact of caring on the QoL and the physical and mental health of informal caregivers [15,16]. They may experience anxiety, depression [17,18], sleep disturbance [19], and a gradual reduction in leisure and social opportunities [20].

Risk factors for informal caregivers’ mental well-being and QoL deterioration include the level of dependency of the person being cared for, dementia, the time spent in caregiving during the week, the lack of training in caregiving and information about available services in the community, and lack of home health care services. Living with an older person is often a risk factor for the health of informal caregivers [21]. Similarly, living in close proximity to the older individual can be both a protective factor and a driver of worsening the QoL of informal caregivers.

The COVID-19 pandemic has contributed to worsening the health outcomes for both informal caregivers and older individuals in need of long-term care. Physical distancing, often translated into social distancing, worsened QoL [22]; accelerated the cognitive decline of many older individuals [23,24]—especially these affected by mild cognitive impairment and dementia [25]—or suffering from neurodegenerative diseases, such as Parkinson’s disease [22,26]; and increased behavioral disturbance in those with various forms of dementia. In addition, during the pandemic, older individuals who received informal care were more likely to be infected than those who did not receive such care, and those with lower incomes had a greater risk of contracting the virus than those with higher incomes [27]. Furthermore, older people with four or more chronic conditions experienced increased loneliness, and females were more likely to have increased anxiety and insomnia [28].

Similarly, the mental well-being and QoL of informal caregivers deteriorated [29] as they were deprived of most public and private support, due to the disruption of semi-residential and home-based care services. The reduced support resulted in an increased burden for them [30].

Although the impact of the pandemic has been widely investigated in the literature, there are few studies that reached the same sample both before and after the outbreak of the pandemic. Thus, there is a lack of evidence on the circumstances that accelerated or mitigated the deterioration in QoL and the mental well-being of community-dwelling older people with long-term care needs, hereafter referred to as older care recipients (OCRs), and their informal caregivers, at the onset of the pandemic. This study aims to contribute to filling the knowledge gap in this regard by conducting a pre/post COVID-19 pandemic comparison of the risk and protective factors for the deterioration of mental well-being and QoL among OCRs and their informal caregivers living in the Marche region, one of the Italian regions with the highest reported life expectancy. As informal caregivers in this study were mainly close relatives, such as spouses or sons/daughters, we refer to them as family caregivers (FCGs) rather than the generic term of informal caregivers (including neighbors and friends).

## 2. Methods

### 2.1. Study Design

This study aims to highlight the predictors of mental well-being and QoL for FCGs and OCRs during the pandemic. We hypothesized that the QoL and mental well-being of FCGs and/or OCRs might be influenced by living alone and feeling safe in one’s own home, the level of fatigue due to caregiving, the satisfaction and the sense of self-efficacy in caring for a family member, financial issues, and the feeling of being respected, heard and listened to, and understood by healthcare professionals before and during the outbreak—a period characterized by the shortage of public and private healthcare services.

Based on a model used in the past for the development of the EUROFAMCARE survey [31], this study includes baseline and follow-up data collected through a survey carried out in the Marche region (in central Italy) to investigate the use of social health services by OCRs and their effectiveness, using two questionnaires: one addressed to the older person with a disability and the other to the main informal caregiver.

The first data collection took place in autumn 2019/winter 2020 and involved 450 caregiver-care recipient dyads from 13 health care districts in the Marche region. Due to the beginning of the COVID-19 health emergency, data collection was interrupted in early March 2020. In the summer of 2020, in the timeframe between the first and the second pandemic waves, OCRs and FCGs were contacted again by phone and were asked to answer some questions to investigate the impact of the pandemic on different domains of life, e.g., mental well-being, QoL, physical health, social contacts, and family relationships. At the follow-up, the number of dyads decreased to 269 because the general climate of concern that characterized the pandemic period demotivated people to respond to telephone interviews. Finally, 239 dyads were included in this study because they completed both the baseline and the follow-up questionnaires.

### 2.2. Respondents’ Recruitment and Inclusion Criteria

Subjects were recruited through the regional pensioners’ trade unions, whose staff also administered the questionnaire after being trained. Written consent was obtained from respondents and all responses were collected anonymously, in compliance with the EU Regulation No. 679 of the European Parliament and of the Council of 27 April 2016 and the Declaration of Helsinki (2013).

Individuals were included and considered eligible to participate in the study if they (a) signed the informed consent form and volunteered to participate in the study; (b1) received the State Care Allowance (“*Indennità di Accompagnamento*” in Italian), a cash benefit provided by the government to people with a severe level of disability or (b2) reported a score of less than 9 on a 12-item scale measuring their level of autonomy in Activities of Daily Living (ADL) and Instrumental Activities of Daily Living (IADL), based on the Barthel index [32,33], whose score ranges from 0 to 99, with scores of 0–20 indicating total dependence and 91–99 slight dependence. Thus, the higher the score, the greater the autonomy of the care recipient; and (c) were 75 years of age or older.

Each trade union selected participants for this study from families who had applied for the State Care Allowance or other LTC benefits through their patronage services.

Once eligible, OCRs were asked to identify their primary FCGs. Where family members were not available, privately paid home-based care workers (mostly live-in) were included as the second component of the dyad. In more than half of the cases (137 out of 236 dyads), the care recipient was unable to complete the questionnaire alone, and the primary caregiver completed most or all of the survey as a proxy for all questions that did not require a subjective response [34,35,36].

To ensure the representativeness of the sample, the proportion of people aged over 75 in each district was estimated using data from the Italian National Statistics Institute (ISTAT) and then stratified by age (75–79 years, 80–84 years, 85 years and over) and by gender. Recruitment problems in some areas have led to a slight over-representation of individuals in some districts and a slight under-representation in others. However, overall representativeness was generally maintained, with deviations in some regional areas being compensated for by others.

### 2.3. Data Collection Tools and Outcome Measures

At baseline, the questionnaire for OCRs included questions on general health conditions, level of dependency, the use of public and private LTC services, informal support networks, social inclusion, and QoL. The questionnaire for FCGs covered care burden, informal support, work–life balance, and the use of public and private LTC services. The Carers of Older People in Europe (COPE) Index was also calculated based on 15 items of the questionnaire [37].

At the follow-up, OCRs and FCGs were asked about how their lives had changed during the pandemic.

The analysis presented here uses data from the baseline and follow-up studies collected for both OCRs and FCGs. All baseline data were included in this analysis. In the follow-up study, the data looked at the mental well-being and QoL of care recipients and family caregivers to understand how their situation had changed compared to before the pandemic. QoL and mental well-being of FCGs and OCRs are the four main outcomes of this study. They were measured with the question “Has your situation changed because of the pandemic?” (with three possible answers: it remained unchanged, it improved, or it worsened) dichotomized into “it worsened” vs. “it improved/unchanged”.

Some factors that might influence the outcome variables were also considered. Firstly, the OCRs’ sense of security in their home was assessed with the question “Do you feel safe in your home?” with a range comprising four possible answers from “1 = very safe” to “4 = very unsafe”; the living conditions were tested with the question “Who do you live with?”; the overall health self-evaluation was assessed through the question “In general how do you rate your health?” (with five possible answers from “1 = excellent” to “5 = bad”).

In addition, FCGs’ feelings about their own caregiving abilities were assessed with the question “Do you feel insecure about what to do for your family member?” with five possible answers from “0 = never” to “4 = almost always”; caregiving fatigue was assessed with the question “Do you feel fatigued when caring for your older person?” with five possible answers from “0 = never” to “4 = almost always”; the importance of healthcare knowledge was tested with the question “How important is it that healthcare professionals (e.g., nurses) have the necessary skills and training to properly care for your family member?” with five possible answers from “0 = not at all” to “4 = extremely”; the family caregivers’ physical health condition was assessed with the question “In general, how do you rate your health?” (with five possible answers from “1 = excellent” to “5 = bad”). Furthermore, FCGs’ opinions on the following issues were tested through ad hoc close multiple-choice closed questions to assess the importance they provided to the fact that their views and opinions were heard and listened to by healthcare professionals (0 = not at all; 4 = extremely); that both their needs and those of the older person were addressed by the services (0 = not at all; 4 = extremely); that professionals treated them and older people with dignity and respect (0 = not at all; 4 = extremely); and whether they felt less able to carry out their role as an assistant (1 = never; 4 = always).

### 2.4. Analysis

Data were expressed as mean and standard deviation, or as number and percentage, depending on the nature of the variables. For all pre-pandemic questionnaire sections, comparisons were made between the worsened group and the improved/unchanged group using chi squared tests and *t*-tests for categorical and continuous variables, respectively. Binary logistic regression models were built to determine predictors using a backward stepwise method, including all statistically significant independent variables from the univariate analysis included, and then by removing them one by one until the best model fit was obtained. For this analysis, some covariates had to be removed because they had a large confidence interval, which could indicate that the sample was not a good representation of the population. Statistical significance was set at *p* < 0.05. All statistical analyses were performed using SPSS version 24 (SPSS Inc., Chicago, IL, USA).

### 2.5. Ethics

This study was submitted for approval to the ethics committee of the National Institute of Health and Science on Ageing (INRCA), Italy. Approval was not deemed necessary, as the study did not involve clinical patients.

## 3. Results

### 3.1. Sample Description

The complete study (baseline and follow-up) includes a sample of 265 OCR-FCG dyads. Between baseline and follow-up, 26 OCRs (9.8%) died, leaving a sample of 239 dyads. The majority of OCRs (71.1%) and FCGs (65.8%) were women. The mean age of participants was 86.1 years for the former and 63.5 for the latter. Regarding the availability of data on the 4 outcome variables, there are, respectively, 1 missing for OCRs and 27 missing for FCGs, corresponding to the changes in well-being. Regarding the changes in QoL, there are, respectively, 3 missing for OCRs and 28 missing for FCGs.

### 3.2. Family Caregivers and Older Care Recipients’ Self-Reported Levels of QoL and Mental Well-Being

Table 1a,b provide an overview of OCRs’ and FCGs’ perceived mental well-being and QoL, before (at baseline) and after the pandemic (at follow-up).

FCGs who reported a worse mental well-being after the pandemic felt less safe at home, more tired caring for the care recipient, and more insecure about their caregiving abilities. They were also less likely to consider whether healthcare professionals had the necessary skills and training after the pandemic than the FCGs who reported the same or better mental well-being (respectively 1.8 vs. 1.7; *p* = 0.050; 2.2 vs. 1.8; *p* = 0.029; 1.6 vs. 1.3; *p* = 0.041; 3.3 vs. 3.5; *p* = 0.041).

OCRs who reported a worse mental well-being after the pandemic were the same ones who had woken up less refreshed and rested in the 2 weeks before the pandemic, compared to older people who maintained or increased their mental well-being (respectively 1.3 vs. 1.9; *p* = 0.017) after the pandemic.

FCGs who reported a worse QoL after the pandemic were less likely to believe that it was important for healthcare professionals to have the necessary skills and training (3.2 vs. 3.5; *p* = 0.033) than the FCGs who reported the same or increased QoL, and that their views and opinions were heard and listened to by healthcare professionals (2.8 vs. 3.1; *p* = 0.002). Moreover, they believed that both their needs and those of the older person were met by the services (3.0 vs. 3.3; *p* = 0.009) and that professionals treated them and older people with dignity and respect (3.4 vs. 3.7; *p* = 0.007 for the elderly; 3.2 vs. 3.5; *p* = 0.008 for themselves. Finally, they felt less able to fulfil their role as an assistant (3.1 vs. 3.4; *p* = 0.012).

OCRs who reported a worse QoL after the pandemic rated their overall health as worse and reported living with fewer people compared to older people with maintained or improved QoL (respectively % of excellent 5.7 vs. 19.5; *p* = 0.032; mean of people living together 2.3 vs. 2.6; *p* = 0.028).

OCRs cared for by the FCGs who reported a worse mental well-being after the pandemic were characterized by a higher percentage (pre-pandemic) of difficulties with different expenses (rent, bills, mortgage, or other loans) during the last year, a lower level of autonomy in climbing stairs, and a higher Barthel score (≥91), compared to the FCGs who had maintained or increased their mental well-being (respectively 42.9% vs. 21.4%; *p* = 0.001; 0.3 vs. 0.5; *p* = 0.027; 70.4 vs. 55.4; *p* = 0.032).

OCRs cared for by the FCGs who reported a worse QoL after the pandemic had woken up less refreshed and rested in the two weeks before the pandemic and rated their QoL worse than the FCGS who maintained or improved their QoL (1.1 vs. 1.9; *p* = 0.002 and 3.2 vs. 2.8; *p* = 0.002, respectively).

FCGs who reported a worse QoL after the pandemic were less able to perform their role of being an assistant before the pandemic. They felt less supported by their friends/neighbors and family, had a poorer relationship with the care recipient, and felt less supported in their caring role in general, compared to the FCGs who had maintained or improved their QoL (respectively 3.0 vs. 3.4; *p* = 0.001; 1.9 vs. 2.2; *p* = 0.046; 3.2 vs. 3.5; *p* = 0.013; 3.6 vs. 3.8; *p* = 0.036; 2.7 vs. 2.9; *p* = 0.024).

FCGs who reported a worse QoL after the pandemic felt less cheerful and in a good mood and less active or full of energy (in the two weeks before the pandemic), compared to the FCGs who had maintained or increased their QoL (respectively 2.1 vs. 2.5; *p* = 0.038; 2.0 vs. 2.6; *p* = 0.011).

For FCGs who reported a worse QoL after the pandemic, it was less important (before the pandemic) that healthcare professionals treated them and the care recipient with dignity and respect, that their views and opinions were heard and listened to, and that care took into consideration both their needs and those of the care recipient, compared to the FCGs who had maintained or improved their QoL (respectively 3.5 vs. 3.7; *p* = 0.019 and 3.2 vs. 3.5; *p* = 0.005; 2.8 vs. 3.2; *p* = 0.002; 3.0 vs. 3.4; *p* < 0.001).

Finally, FCGs who reported a worse QoL after the pandemic had a lower COPE Index (before the pandemic), compared to those who maintained or improved their QoL (respectively 43.3 vs. 45.3; *p* = 0.018).

### 3.3. Factors Associated to Mental-Being and QoL

Table 2 shows four binary logistic analyses of the predictors associated with worsening in mental well-being and QoL in OCRs and FCGs after the pandemic, adjusting for age, gender, and other confounders in the model.

During the pandemic, the importance that FCGs considered regarding the skills and training of healthcare professionals’ buffered the deterioration of OCRs’ mental-wellbeing during the pandemic (OR = 0.569; 95% CI: 0.365–0.886). Similarly, the importance that FCGs provided to healthcare professionals’ attention to both their needs and those of older people was a protective factor for the latter’s QoL (OR = 0.526; 95% CI: 0.338–0.817).

Older care recipient’s difficulty in paying rent, bills, mortgage, or other loans may have worsened the family caregivers’ mental well-being (OR = 2.593; 95% CI: 1.370–4.911). Family caregivers’ age was a risk factor for worsening the QoL (OR = 1.039; 95% CI: 1.001–1.079): the older the age, the higher the risk. Also, a lower perceived older care recipient’s QoL before the pandemic was a risk factor for the QoL worsening in the family caregivers during the pandemic (OR = 1.928; 95% CI: 1.061–3.503). Finally, family support of the family caregivers before the pandemic was a protective factor for worsening the QoL during the pandemic (OR = 0.580; 95% CI: 0.349–0.963).

## 4. Discussion

This study provides information on the risk and protective factors for the changes in mental well-being and QoL in 239 dyads of OCRs and FCGs who responded to a questionnaire administered before and after the first wave of the pandemic that hit Italy in spring 2020.

In line with the literature, in the Marche region, where the study was carried out, the high demand for long-term care by numerous older residents with multiple chronic diseases, was mainly met by family caregivers, reflecting the family-oriented welfare system that characterizes the country [12,13,14]. Moreover, the analysis confirms the enormous negative impact of the pandemic (with the interruption of the public and private care services and physical distancing measures) on the QoL of OCRs [22] and on the mental well-being and QoL of FCGs [29,30], in addition to the negative impact that is usually experienced in non-pandemic periods [15,16,17,18,19,20].

The financial difficulties of older care recipients contributed to a deterioration of the mental well-being of family caregivers, probably because part of their mental workload was focused on how to manage the care recipient’s expenditure regarding LTC services. Thus, low income not only affected the QoL of OCRs who became infected [27], but it might have also affected the mental well-being of FCGs.

On the one hand, this study highlighted that the belief of care recipients that their OCRs might be at risk in their own home during the pandemic negatively affected the mental well-being of family caregivers. On the other hand, living alone and reporting poor health were the main causes of worsening the QoL among older care recipients.

These findings call for policy recognition of older people’s safety, social isolation, and loneliness as public health issues that the pandemic has highlighted and that need to be addressed through more effective and specific interventions.

Moreover, a low recognition of the importance of trust, a cooperative relationship, and clear and effective communication with healthcare professionals had a negative impact on FCGs’ QoL. This finding suggests that there is a need for training to focus on communication and for it to target both informal and formal caregivers, and it also confirms the importance of integrating formal and informal care for older people with LTC needs. Such integration is still not fully addressed in Italy, where there are no clear regulations on how family members can cooperate with care professionals, both at home and in residential care facilities. This was particularly not addressed during the pandemic, when many services were disrupted, cancelled, or postponed and many sons/daughters and spouses had to take over or increase their role as the main caregiver [38,39] without any form of cooperation.

Furthermore, the impact of the pandemic was greater for family caregivers with low coping skills and low levels of self-confidence, thus highlighting the need for psycho-educational interventions to support them.

These findings confirm the need for ongoing training for family/informal caregivers to improve access to services [40], increase well-being [41,42] and self-efficacy [43], and improve their QoL [44].

Such policies and measures are still lacking in the Italian legislation. Even the recently adopted national Law n. 33/2023, which aims to reform the Italian Long-Term Care system, on the basis of a common definition of dependency, lacks an explicit emphasis on strengthening the support for informal caregivers. In fact, the current reform offers only cursory provisions aimed at generally enhancing caregivers’ living conditions. These provisions include a redefinition of the rules, improved certification of the professional skills acquired by caregivers in their caregiving role, and a greater involvement of their representatives in the planning of health and social care services.

This study is not without limitations. The main one is the use of proxies for OCRs with a cognitive decline, derived from the responses of family caregivers. As suggested by Roydhouse et al. [35], it would have been more methodologically correct to test the proxy–patient concordance using validated instruments adapted for people with dementia and measures developed specifically for proxies, and then to compare the consistency and reliability of the responses.

Another limitation arises from the fact that this study was not designed from the outset with a double survey and that the second survey was conducted in the context of a health emergency. Therefore, although the researchers did their best to ensure that the follow-up questionnaire contained questions that were consistent with the baseline questionnaire, they still had to ask only essential questions, as data collection was performed via telephone due to the rules on physical distancing.

Another limitation is the lack of measurement of the pandemic’s impact and the reduction in care services on the cognitive abilities and psychological well-being of older people and family caregivers using psychometric scales. These aspects were assessed by a question that asked for the perceived impact of the respondents and therefore provided subjective and not generalizable data. However, such scales should be administered F2F by healthcare professionals (e.g., neuropsychologists), and the physical distance rules for the containment of contagion did not allow respondents to be reached in their homes or in healthcare facilities.

Finally, this study monitored the general health status of the respondents before and after the first wave of the pandemic, but it is not possible to verify whether the deterioration in their health was due to the normal ageing process or to the cumulative effect of the latter and the physical and social restrictions of the pandemic.

## 5. Conclusions

The economic uncertainty, social insecurity, fear for one’s health, and relational poverty that characterized the pandemic period worsened the QoL and mental well-being of both older persons with LTC needs and family caregivers. The deterioration of the general life condition of older care recipients and family caregivers may be counteracted by attentive healthcare professionals, a supportive family environment, and economic support that can make both the caregiver and the older care recipient feeling safe, well cared for, and understood, because part of a care environment characterized by stable care relationships based on a strong union between formal and informal care.

These aspects should be considered not only in the presence of health emergencies, but whenever care interventions are planned for community-dwelling older people with LTC needs.

## Figures and Tables

**Table 1 healthcare-12-00383-t001:** (**a**) Older care recipients’ (OCRs’) changes in mental well-being and QoL after the pandemic. (**b**) Family caregivers’ (FCGs’) changes in mental well-being and QoL after the pandemic.

**(a)**			
**Older Care Recipients’ Mental Well-Being**			
	**Improvement/Stability after Pandemic (n = 134)**	**Worsening after Pandemic (n = 104)**	** *p* **
FCGs			
Older adults’ safety in their home (1 = very safe; 4 = very unsure)	1.7 ± 0.7	1.8 ± 0.6	0.050
Fatigue when caring for an older adult (0 = never; 4 = almost always)	1.8 ± 1.2	2.2 ± 1.1	0.029
Uncertain about what to do for their own older adult (0 = never; 4 = almost always)	1.3 ± 1.1	1.6 ± 1.2	0.041
Importance of caregiver skills and knowledge (0 = not at all; 4 = extremely)	3.5 ± 0.8	3.3 ± 0.9	0.041
OCRs			
Frequency of waking up refreshed and rested (0 = never; 5 = always)	1.9 ± 1.6	1.3 ± 1.3	0.017
**Older care recipients’ QoL**			
	**Improvement/stability after pandemic (n = 138)**	**Worsening after pandemic (n = 98)**	** *p* **
FCGs			
Frequency of satisfaction in carrying out the role of assistant (1 = never; 4 = always)	3.4 ± 0.6	3.1 ± 0.7	0.012
Importance of caregiver skills and knowledge (0 = not at all; 4 = extremely)	3.5 ± 0.7	3.2 ± 1.0	0.033
Importance of caregivers treating older adults with dignity and respect (0 = not at all; 4 = extremely)	3.7 ± 0.6	3.4 ± 1.0	0.007
Importance of health professionals treating caregivers with dignity and respect (0 = not at all; 4 = extremely)	3.5 ± 0.8	3.2 ± 1.0	0.008
Importance of caregiver’s views and opinions being heard and listened to (0 = not at all; 4 = extremely)	3.1 ± 0.8	2.8 ± 1.0	0.002
Importance of assistance taking into account the needs of caregivers and older adult (0 = not at all; 4 = extremely)	3.3 ± 0.8	3.0 ± 0.9	0.009
OCRs			
Perceived health (excellent)	17 (19.5)	4 (5.7)	0.032
Total number of people living together (including older adults)	2.6 ± 1.3	2.3 ± 1.0	0.028
(**b**)			
**Family Caregivers’ mental well-being**			
	**Improvement/stability after pandemic (n = 113)**	**Worsening after pandemic (n = 99)**	** *p* **
OCRs			
Difficulty with the following expenses (if applicable) for rent, utility bills, mortgage, or other loans in the last year (yes)	24 (21.4)	42 (42.9)	0.001
Autonomy in climbing the stairs (0 = not able; 1 = with a little help; 2 = without help)	0.5 ± 0.6	0.3 ± 0.5	0.027
Barthel Scale ≥ 91	62 (55.4)	69 (70.4)	0.032
**Family Caregivers’ QoL**			
	**Improvement/stability after pandemic (n = 125)**	**Worsening after pandemic (n = 86)**	** *p* **
FCGs			
Frequency of satisfaction in carrying out the role of assistant (1 = never; 4 = always)	3.4 ± 0.6	3.0 ± 0.7	0.001
Frequency of support from friends/neighbors (1 = never; 4 = always)	2.2 ± 1.0	1.9 ± 0.9	0.046
Frequency of support from family (1 = never; 4 = always)	3.5 ± 0.8	3.2 ± 0.9	0.013
Relationship with older adults is good (1 = never; 4 = always)	3.8 ± 0.5	3.6 ± 0.6	0.036
Frequency of global support for the role of aid (1 = never; 4 = always)	2.9 ± 0.8	2.7 ± 0.8	0.024
Frequency of feeling happy and in a good mood in the last 2 weeks (0 = never; 5 = always)	2.5 ± 1.3	2.1 ± 1.3	0.038
Frequency of feeling active and full of energy in the last 2 weeks (0 = never; 5 = always)	2.6 ± 1.5	2.0 ± 1.4	0.011
Importance of caregivers treating older care recipients with dignity and respect (0 = not at all; 4 = extremely)	3.7 ± 0.5	3.5 ± 0.9	0.019
Importance of health professionals treating caregivers with dignity and respect (0 = not at all; 4 = extremely)	3.5 ± 0.7	3.2 ± 0.9	0.005
Importance of caregiver’s views and opinions being heard and listened to (0 = not at all; 4 = extremely)	3.2 ± 0.7	2.8 ± 0.9	0.002
Importance of assistance taking into accounts the needs of caregivers and older care recipients (0 = not at all; 4 = extremely)	3.4 ± 0.8	3.0 ± 0.9	<0.001
COPE *	45.3 ± 6.1	43.3 ± 5.2	0.018
OCRs			
Frequency of waking up refreshed and rested (0 = never; 5 = always)	1.9 ± 1.6	1.1 ± 1.2	0.002
Perceived QoL in the last two weeks (1 = very good, 5 = very bad)	2.8± 0.7	3.2 ± 0.6	0.002

* COPE: Carers of Older People in Europe Index.

**Table 2 healthcare-12-00383-t002:** Predictors of mental well-being deterioration and QoL worsening in older care recipients (OCRs) and family caregivers (FCGs) after the pandemic.

	**OR**	**95% CI for EXP(B)**		** *p* **
		**Lower**	**Upper**	
Deterioration of older care recipient’s mental well-being				
OCRs				
Age	0.953	0.897	1.012	0.119
Gender (female)	0.744	0.329	1.682	0.477
Frequency of waking up refreshed and rested	0.848	0.651	1.105	0.223
FCGs				
Older adults’ safety in their home	1.469	0.837	2.577	0.180
Fatigue in caring for older adults	1.175	0.859	1.606	0.314
Importance of caregiver skills and knowledge	0.569	0.365	0.886	0.013
Worsening of older care recipient’s QoL				
OCRs				
Age	1.001	0.943	1.062	0.979
Gender (female)	0.895	0.369	2.168	0.805
Perceived health (excellent)	1.808	0.458	7.130	0.398
Total number of people living together (including older adults)	0.742	0.505	1.091	0.129
FCGs				
Frequency of satisfaction in carrying out the role of assistant	0.657	0.370	1.165	0.151
Importance of healthcare professionals taking into account the needs of caregivers and older care recipients	0.526	0.338	0.817	0.004
Deterioration of family caregiver’s mental well-being				
FCGs				
Age	1.013	0.986	1.041	0.356
Gender (female)	0.724	0.394	1.333	0.300
OCRs				
Difficulty with the following expenses (if applicable) for rent, utility bills, mortgage, or other loans in the past year	2.593	1.370	4.911	0.003
Autonomy in climbing the stairs	0.664	0.355	1.242	0.200
BARTHEL SCALE (Ref. 0–20)				0.227
21–60	1.434	0.651	3.159	
61–90	0.550	0.175	1.723	
≥91	1.298	0.076	22.194	
Worsening of family caregiver’s QoL				
FCGs				
Age	1.039	1.001	1.079	0.044
Gender (female)	0.907	0.409	2.011	0.811
Frequency of support received from friends/neighbors	0.797	0.541	1.175	0.252
Frequency of support received from family	0.580	0.349	0.963	0.035
Relationship with the older care recipient (good)	0.576	0.286	1.160	0.122
OCRs				
Perceived QoL in the last two weeks	1.928	1.061	3.503	0.031

## Data Availability

The data presented in this study are available upon request from the corresponding author. The data are not publicly available because they are still object of ongoing analyses.

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
