# Peer review of "Predictors of Deterioration in Mental Well-Being and Quality of Life among Family Caregivers and Older People with Long-Term Care Needs during the COVID-19 Pandemic"

_healthcare, 2024, doi:10.3390/healthcare12030383_

Round 1
Reviewer 1 Report
Comments and Suggestions for Authors
Very interesting and well written paper. I do not see the issue that the care recipients aged during the time period. Might the results be also affected by the aging, and increased disabilities due to the age-related deterioration of chronic issues, during the two periods?
Author Response
Dear reviewer,
We Authors would like to thank you for your useful and constructive suggestions that contributed to improve the quality of the manuscript.
Below we reported your comments/suggestions, followed by our answers in bold.
Best regards,
The Authors
Reviewer 1
Very interesting and well written paper. I do not see the issue that the care recipients aged during the time period.
Might the results be also affected by the aging, and increased disabilities due to the age-related deterioration of chronic issues, during the two periods?
AUTHORS’ ANSWER:
Between the first and the second data collection wave there were about six months, during which, due to the physical distancing measures imposed by the pandemic, older people stayed most “at home”. The study monitored the general health status of the respondents through the questionnaire but it is not possible to verify if the health deterioration depends on the normal ageing process or on the cumulative effect of the latter and the pandemic physical and social restrictions. We added this aspect to the study limitation, at the end of the Discussion.
Reviewer 2 Report
Comments and Suggestions for Authors
You could relate your findings to previous research, providing more detailed analysis of how your obtained results compare to those used in other studies. I recommend that you include a reference to a model of personal care and residential center/home.
Author Response
Dear reviewer,
We Authors would like to thank you for your useful and constructive suggestions that contributed to improve the quality of the manuscript.
Below we reported your comments/suggestions, followed by our answers in bold.
Best regards,
The Authors
-You could relate your findings to previous research, providing more detailed analysis of how your obtained results compare to those used in other studies.
AUTHORS’ ANSWER: In the Discussion, we added a paragraph where we relate the results to the literature.
-I recommend that you include a reference to a model of personal care and residential center/home.
AUTHORS’ ANSWER: The study is based on the same research model used in the EUROFAMCARE survey. We explained this in the manuscript text and added a reference.
Reviewer 3 Report
Comments and Suggestions for Authors
The quality of the English language is good.
Author Response
Dear reviewer,
We Authors would like to thank you for your useful and constructive suggestions that contributed to improve the quality of the manuscript.
Below we reported your comments/suggestions, followed by our answers in bold.
Best regards,
The Authors
In order to publish the article, I recommend the authors to make the following adjustments:
- At lines 106 and 198, it is better not to start the sentence with a number. For example, “Finally, 239 dyads were considered …”, “During the study, 26 subjects in need of care …”.
AUTHORS’ ANSWER: We addressed this suggestion.
- In sub-section 2.3, at line 141, please shortly explain what COPE means. It can be evident for you, as authors, but for the uninitiated public, it is the first place in the manuscript that it is mentioned, but not explained.
AUTHORS’ ANSWER: We addressed this suggestion
- In the same sub-section (2.3), there are more places where you should the correct the number of the answering categories, from four to five, as follows:
- at lines 160 and 170, the categories from 1 to 5 means five, not four possible answers;
- at lines 163 and 168, the categories from 0 to 4 means five, not four possible answers.
AUTHORS’ ANSWER: All mistakes were amended.
- Please check are erase the extra spaces from line 167
– after the parenthesis (e.g. nurses); line 201 (before the sentence starting with “In …”);
lines 237-238 (erase the space before the dot from the end of the sentence;
line 251(before the sentence “For family caregivers …”).
AUTHORS’ ANSWER: All extra spaces were erased.
- In the paragraphs comprised between lines 155-177, please edit uniformly the spaces of the texts around the sign “=”. In some cases, before or after this sign there is a space, but in other cases, there is no space before or after this sign!
AUTHORS’ ANSWER: We edited it uniformly.
- Related to the same two paragraphs, usually, when examples of questions from the data collection instruments are given, the authors use either the question marks or the italic characters, and not both. I suggest choosing only one of these two options.
AUTHORS’ ANSWER: We chose to withdraw the Italics and maintain the question marks.
- At lines 171-172, did you mean: “ad hoc closed 171 multiple-choice questions”?
- At lines 172-173, did you mean: “that the importance of their views and opinions were heard by healthcare professionals”?
- At lines 171-177, for a smoother reading, please separate through the sign “;”, the four topics assessed through ad hoc closed multiple-choice questions.
AUTHORS’ ANSWER: To address the three points above we rephrased the sentence from line 172 as follows: Furthermore, family caregivers’ opinion about the following topics were checked through ad hoc close multiple-choice questions assessing the importance they gave to the fact: that their views and opinions were heard by healthcare professionals (0=not at all; 4=extremely); that both their needs and those of the older person were addressed by the services (0=not at all; 4=extremely); that professionals treated them and older people with dignity and respect (0=not at all; 4=extremely); and whether they felt less able to carry out their role of assistant (1=never; 4=always).
- At line 206, please renumber sub-section 3.1. Family Caregivers and Older People’ SelfReported Levels of Qol and Mental Well-Being with 3.2.
Also, at line 260, renumber subsection 3.2. Factors associated to mental-being and QoL with 3.3.
AUTHORS’ ANSWER: We re-numbered the two sub-sections
- At line 235 did you mean “a higher percentage of Barthel score”?
AUTHORS’ ANSWER: We mean a higher Barthel score. We amended the text.
We also added the explanation of the Barthel scale score in the sub-paragraph 2.2 as follows: “[…] based on the Barthel index [31, 32], whose score ranges between 0 and 99, where the scores 0-20 indicate total dependency and 91-99 slight dependency. Thus, the higher the score, the greater the autonomy of the care recipient”.
- At line 265, at the end of the title of Table 2, please erase the dot!
AUTHORS’ ANSWER: We removed the dot after the title of Table 2 and we also harmonised the tables title style over the manuscript.
- Related to the information provided in Table 2, I suggest the authors specify more clearly which is the reference category for each predictor (e.g. for gender and financial difficulty is clear, but it is not the same for other explanatory factors)! In addition, please insert in this table the value of p for each predictor, too!
AUTHORS’ ANSWER: In Table 2 the reference category for each categorical predictor is shown into the brackets. The other factors are calculated in continuous form and their directions are shown in Table 1. We added p-values for each predictor.
- In the Discussion section, the authors should introduce at least one paragraph in which they relate their results with those of similar studies. Also, the authors should expose the main limitations of their research!
AUTHORS’ ANSWER: In the Discussion, we added a paragraph where we relate the results to the literature and discuss the study limitations.
Reviewer 4 Report
Comments and Suggestions for Authors
Thank you for the opportunity to review the article "Predictors of
Decline in Mental Well-being and Quality of Life Among Family Caregivers
and Older Care Recipients During the COVID-19 Pandemic."
The study underscores the significance of confronting challenges faced
by family caregivers, especially during the pandemic, and highlights the
need for developing comprehensive policies and support measures to
improve the well-being of both caregivers and older individuals
requiring long-term care. Lack of cooperation was particularly evident
during the pandemic when many services were disrupted or canceled,
compelling family members to assume or increase their roles as primary
caregivers without adequate support. The study focused on analyzing
factors influencing changes in the mental well-being and quality of life
of long-term care recipients and their family caregivers before and
after the Covid-19 pandemic.
My remarks aim to specify and clarify several issues:
1. How the authors assessed the mental health of the respondents;
2. Doubtful responses provided by caregivers instead of patients;
3. Lack of data on the number of individuals who passed away during the
study period, lines 198-199;
4. What skills were expected from nursing staff or the type of support
required.
Author Response
Dear reviewer,
We Authors would like to thank you for your useful and constructive suggestions that contributed to improve the quality of the manuscript.
Below we reported your comments/suggestions, followed by our answers in bold.
Best regards,
The Authors
Thank you for the opportunity to review the article "Predictors of Decline in Mental Well-being and Quality of Life Among Family Caregivers and Older Care Recipients During the COVID-19 Pandemic."
The study underscores the significance of confronting challenges facedby family caregivers, especially during the pandemic, and highlights the need for developing comprehensive policies and support measures to improve the well-being of both caregivers and older individuals requiring long-term care. Lack of cooperation was particularly evident during the pandemic when many services were disrupted or canceled, compelling family members to assume or increase their roles as primary caregivers without adequate support. The study focused on analysing factors influencing changes in the mental well-being and quality of life of long-term care recipients and their family caregivers before and after the Covid-19 pandemic.
My remarks aim to specify and clarify several issues:
1. How the authors assessed the mental health of the respondents;
AUTHORS’ ANSWER: Mental health, and also quality of life of the respondents were measured by the question “Has your situation changed due to the pandemic?” with three possible answers (it remained unchanged, it improved, or it worsened) that were dichotomized into “it worsened” vs “it improved/unchanged” (please see “2.3. Data Collection Tools and Outcome Measures”).
Doubtful responses provided by caregivers instead of patients;
AUTHORS’ ANSWER: The answers of family caregivers on the health condition of the older care recipients were treated as proxy of the latter as explained in the following sentence in the manuscript: “In more than half of the cases (137 out of 236 dyads), the person in need of care was unable to complete the questionnaire on their own, and therefore the primary caregiver completed most, or all of the survey as a proxy for all questions that did not require a subjective response”.
This method is quite common especially in surveys and studies involving older people with dementia, because the cognitive decline of patients with dementia makes it impossible to answer questions on subjective topics, such as for example the health related QoL (e.g. a) Albert SM, Del Castillo-Castaneda C, Sano M, Jacobs DM, Marder K, Bell K, Bylsma F, Lafleche G, Brandt J, Albert M, et al: Quality of life in patients with Alzheimer's disease as reported by patient proxies. J Am Geriatr Soc. 1996, 44 (11): 1342-1347; b) Krabbe, P., Schölzel-Dorenbos, C. J. M., & Rikkert, M. G. O. (2010). Quality of life in dementia patients and their proxies; a narrative review of the concept and measurement scales. In: Handbook of Disease Burdens and Quality of Life Measures. Handbook of Disease Burdens and Quality of Life Measures, 3671-3689; c) Roydhouse JK, Cohen ML, Eshoj HR, Corsini N, Yucel E, Rutherford C, Wac K, Berrocal A, Lanzi A, Nowinski C, Roberts N, Kassianos AP, Sebille V, King MT, Mercieca-Bebber R; ISOQOL Proxy Task Force and the ISOQOL Board of Directors. The use of proxies and proxy-reported measures: a report of the international society for quality of life research (ISOQOL) proxy task force. Qual Life Res. 2022 Feb;31(2):317-327. doi: 10.1007/s11136-021-02937-8. Epub 2021 Jul 12. PMID: 34254262.)
Thus, reliable and valid informal caregiver (proxy) ratings could prove extremely useful in the field of dementia.
However, using proxy ratings can be a bias.
So, for addressing this comment we:
- added the three references above to the manuscript in the Method paragraph;
- highlighted the possible bias of using proxies in a new section of the Discussion devoted to the study limitations, as suggested by Reviewer n. 3.
Lack of data on the number of individuals who passed away during the study period, lines 198-199;
AUTHORS’ ANSWER: We clarified the number of individuals who passed away during the study period with the following sentence: “The complete (baseline and follow-up) study includes a sample of 265 OCR-FCG dyads. Between baseline and follow-up, 26 OCRs (9.8%) died, leaving a sample of 239 dyads”.
What skills were expected from nursing staff or the type of support required.
AUTHORS’ ANSWER: No nursing staff was involved in this study. In case this comment refers to the trade unions’ staff who administered the questionnaire, they were trained by means of ad hoc training sessions during which all sections of the questionnaire were thoroughly explained and interview simulations carried out, thus giving interviewers the opportunity of gaining a deep understanding of the data collection tool.
Reviewer 5 Report
Comments and Suggestions for Authors
The study is interesting, but I have some difficulties with the analyses and presentation.
The same group of participants in the study is variably refered to as 'older people', 'people in need of care', 'older care recipiens', 'older adults', and 'care recipients'. Only one of these terms should be used consistently in text and tables, suggest 'care receivers'
In the inclusion criteria, if there is a lower age limit, this should be given. In the Sample description, age and gender distribution should be given.
In Table 1, as I understand it, the participants are, for each section of the table, divided in two groups of improvement vs. worsening by one result of the study, and then mean and SD of the two groups are compared for another variable. I would find it more meaningful to compare the mean scores of caregivers before and after the pandemic, and the same for care recipients. The numbers of participants with improvement and worsening respectably could be given separately , preferably in the text.
The text referring to the table has several very long periods of up to 9 lines and a number of variables without a full stop, which makes it difficult to read and get the meaning. A simpler language with less information in each period is recommended.
Comments on the Quality of English Language
The subject and study are intresting, but in my opinion, the analyses should be redone as suggested above
Author Response
Reviewer 5
Dear reviewer,
We Authors would like to thank you for your useful and constructive suggestions that contributed to improve the quality of the manuscript.
Below we reported your comments/suggestions, followed by our answers in bold.
Best regards,
The Authors
-The study is interesting, but I have some difficulties with the analyses and presentation.
-The same group of participants in the study is variably referred to as 'older people', 'people in need of care', 'older care recipients', 'older adults', and 'care recipients'. Only one of these terms should be used consistently in text and tables, suggest 'care receivers'
AUTHORS’ ANSWER: We chose to use “older care recipient” and the abbreviation OCR along the overall manuscript text. We also chose the abbreviation FCGs for family caregivers throughout the manuscript.
-In the inclusion criteria, if there is a lower age limit, this should be given. In the Sample description, age and gender distribution should be given.
AUTHORS’ ANSWER: We added the lower age limit in the inclusion criteria (paragraph “2.2. Respondents’ Recruitment and Inclusion Criteria”). In the sample description we simply talked about age and gender distribution of the sample, divided by groups (OCRs and FCGs).
-In Table 1, as I understand it, the participants are, for each section of the table, divided in two groups of improvement vs. worsening by one result of the study, and then mean and SD of the two groups are compared for another variable. I would find it more meaningful to compare the mean scores of caregivers before and after the pandemic, and the same for care recipients. The numbers of participants with improvement and worsening respectably could be given separately, preferably in the text.
AUTHORS’ ANSWER: Unfortunately, we have no information after the pandemic. In the follow-up database, we only have questions about any status changes from before compared to after the pandemic.
-The text referring to the table has several very long periods of up to 9 lines and a number of variables without a full stop, which makes it difficult to read and get the meaning. A simpler language with less information in each period is recommended.
AUTHORS’ ANSWER: We divided long sentences in two or more, shorter ones.
Round 2
Reviewer 5 Report
Comments and Suggestions for Authors
I accept the authors' response to my previous suggestions as adquate
Author Response
REVIEWER/AUTHORS
Thanks for your excellent work on the original paper and your responses to reviewers. I think there are still issues that they raised that you have not addressed.
1) provide a little more clarity about what the authors expected to find and why.
AUTHORS’ ANSWER: We specified our expectations at the beginning of par 2.1 Study design.
2) provide a more clear rationale for the independent variables considered (such as a theory frame or specific prior findings.)
AUTHORS’ ANSWER: As we declared in the Analysis paragraph, comparisons were conducted between worsened group and improved/unchanged group for all questionnaire sections before pandemic. Therefore all baseline variables were considered as independent and we reported in table 1 only those that were significant.
3) although the sample development section has been updated, it is still not clear how individuals were identified to participate. (Were they selected from program records, clinical records, outreach to a much larger population etc.?
AUTHORS’ ANSWER: We added further details in par 2.2. Respondents’ Recruitment and Inclusion Criteria.
4) the final sample is almost 1/2 of the original cohort. It would be useful to know how they are similar and different from that group on multiple dimensions and what are the implications of this for understanding finding. There is a comparison of the sample to the population of OCRs by age and gender, but not enough is said about the representativeness of the sample.
AUTHORS’ ANSWER: As we declared in the Sample Description paragraph, “the complete (baseline and follow-up) study includes a sample of 265 OCR-FCG dyads. Between baseline and follow-up, 26 OCRs (9.8%) died, leaving a sample of 239 dyads. The majority of OCRs (71.1%) and FCGs (65.8%) were women. The mean age of participants was 86.1 years for the first and 63.5 for the second. In regards to availability of data about the four outcome variables, there are respectively one missing for OCRs and 27 missing for FCGs, corresponding to the changes in well-being. As for changes in QoL, there are respectively three missing for OCRs and 28 missing for FCGs”.
Summarizing, we have four outcomes (see Table 1) with 238, 236, 212 and 211 data available respectively out of 239 dyads. I think that there was a misunderstanding, because calculating their percentages on 239, the final sample is larger than half of the original court (99.5%, 98.7%, 88.7% and 88.2% respectively).
5) Provide or acknowledge as a weakness of the study that there is no psychometric data on the measures of experience with home care and how it impacted QoL.
AUTHORS’ ANSWER: We added this aspect in the part of the Discussion dedicated to the limitation of the study.
6) There are many many bivariate analyses this means that the alpha to reject the null hypothesis needs to be higher to avoid chance observations being considered findings. You could use something like the Bonferroni test for this or similar approach. This may be less of a problem for the multivariate analyses and I would still be suspicious of terms with marginal significance given the number of tests.
AUTHORS’ ANSWER: Thank you for the comment. We were initially thinking about the possibility of multiple correction, but for our study it would have been a too conservative choice. Furthermore, we had a large number of tests, but they were carried out without preplanned hypotheses. So we decided that was better to obtain false significant results in bivariate analyses and further explore the hypotheses in subsequent multivariate analyses, instead of “unfairly neutering” too much evidence by excessively lowering alpha levels, a salso suggested by Rothman KJ. No adjustments are needed for multiple comparisons. Epidemiology. 1990 Jan;1(1):43-6. PMID: 2081237.
7) The discussion mentions that the findings underscore the negative consequences of interutped eldecare services---I did not understand how this conclusion was reached.
AUTHORS’ ANSWER: Thank you, we amended the sentence in “Moreover, the analysis confirms the enormous negative impact of the pandemic, with the interruption of the public and private care services and physical distancing measures, on the QoL of OCRs [22] and on the mental well-being and QoL of FCGs [29] [30], in addition to the negative impact that is usually tested in non-pandemic periods [15-20]”.
8) There needs to be another round of English editing----I noticed incomplete sentences and some word confusions.
AUTHORS’ ANSWER: We have done another proof reading of the paper.
